A horizon scan of future threats and opportunities for pollinators and pollination

Brown Mark J.F. mark.brown@rhul.ac.uk 1
Dicks Lynn V. 2
Paxton Robert J. 3 4
Baldock Katherine C.R. 5 6
Barron Andrew B. 7
Chauzat Marie-Pierre 8
Freitas Breno M. 9
Goulson Dave 10
Jepsen Sarina 11
Kremen Claire 12
Li Jilian 13
Neumann Peter 14
Pattemore David E. 15
Potts Simon G. 16
Schweiger Oliver 17
Seymour Colleen L. 18 19
Stout Jane C. 20
1 School of Biological Sciences, Royal Holloway University of London , Egham , United Kingdom
2 Conservation Science Group, Department of Zoology, University of Cambridge , Cambridge , United Kingdom
3 Institute for Biology, Martin-Luther-University Halle-Wittenberg , Halle , Germany
4 iDiv, German Centre for Integrative Biodiversity Research Halle-Jena-Leipzig , Leipzig , Germany
5 School of Biological Sciences, University of Bristol , Bristol , United Kingdom
6 Cabot Institute, University of Bristol , Bristol , United Kingdom
7 Department of Biological Sciences, Macquarie University , Sydney , Australia
8 European reference laboratory for honeybee health, Unit of honeybee pathology & Unit of coordination and support to surveillance, ANSES , Maisons-Alfort Cedex , France
9 Departamento de Zootecnia, Centro de Ciências Agrárias, Universidade Federal do Ceará , Fortaleza Ceará , Brazil
10 School of Life Sciences, University of Sussex , Falmer , United Kingdom
11 The Xerces Society for Invertebrate Conservation , Portland , OR , United States of America
12 Berkeley Food Institute, Environmental Sciences Policy and Management, University of California Berkeley , Berkeley , CA , United States of America
13 Institute of Apicultural Research, Chinese Academy of Agricultural Sciences , Beijing , China
14 Institute of Bee Health, Vetsuisse Faculty, University of Bern , Bern , Switzerland
15 The New Zealand Institute for Plant & Food Research Limited , Hamilton , New Zealand
16 Centre for Agri-Environmental Research, School of Agriculture, Policy and Development, University of Reading , Reading , United Kingdom
17 Department of Community Ecology, Helmholtz Centre for Environmental Research—UFZ , Halle , Germany
18 South African National Biodiversity Institute, Kirstenbosch Research Centre , Claremont , South Africa
19 Percy FitzPatrick Institute of African Ornithology, DST/NRF Centre of Excellence, Department of Biological Sciences, University of Cape Town , Rondebosch , South Africa
20 Botany, School of Natural Sciences, Trinity College Dublin, the University of Dublin , Dublin , Ireland
Benelli Giovanni
Electronic publication date: 2016 Aug 9
Publication date: 2016
Volume: 4
Electronic Location ID: e2249
Received 2016 Apr 22; Accepted 2016 Jun 22
Copyright: ©2016 Brown et al.
Copyright year: 2016
Copyright holder: Brown et al.
License: This is an open access article distributed under the terms of the Creative Commons Attribution License, which permits unrestricted use, distribution, reproduction and adaptation in any medium and for any purpose provided that it is properly attributed. For attribution, the original author(s), title, publication source (PeerJ) and either DOI or URL of the article must be cited.
License URL: https://creativecommons.org/licenses/by/4.0/

Keywords: Horizon scanning, Pollinator, Pollination, Ecosystem services, Conservation

Funding: Super-B, an EU COST-Action BBSRC BB/N000668/1 NERC NE/K015419/1 NERC Knowledge Exchange Fellowship NE/M006956/1 Macquarie University and USDA 58-5342-3-004F National Council for Scientific and Technological Development-Brazil 305126/2013-0 Chinese National Natural Science Foundation 31572338 Agricultural Science and Technology Innovation Program CAAS-ASTIP-2016-IAR Vinetum Foundation New Zealand’s Ministry of Business, Innovation and Employment C11X1309 DFG Pa 632/10 The Horizon-scanning workshop was supported by Super-B, an EU COST-Action. MJFB was funded by the BBSRC (grant code BB/N000668/1). LVD was funded by the NERC (grant code NE/K015419/1). KCRB was funded by a NERC Knowledge Exchange Fellowship (grant code NE/M006956/1). ABB received funding from Macquarie University and USDA (Grant 58-5342-3-004F). BMF’s participation was supported through the National Council for Scientific and Technological Development-Brazil (No. 305126/2013-0). LJL’s participation was supported by the Chinese National Natural Science Foundation (No. 31572338) and The Agricultural Science and Technology Innovation Program (CAAS-ASTIP-2016-IAR). PN was supported financially by the Vinetum Foundation. DEPs participation was supported through New Zealand’s Ministry of Business, Innovation and Employment contract no. C11X1309. RJP was funded by DFG grant Pa 632/10. The funders had no role in study design, data collection and analysis, decision to publish, or preparation of the manuscript.

==============================
Background. Pollinators, which provide the agriculturally and ecologically essential service of pollination, are under threat at a global scale. Habitat loss and homogenisation, pesticides, parasites and pathogens, invasive species, and climate change have been identified as past and current threats to pollinators. Actions to mitigate these threats, e.g., agri-environment schemes and pesticide-use moratoriums, exist, but have largely been applied post-hoc. However, future sustainability of pollinators and the service they provide requires anticipation of potential threats and opportunities before they occur, enabling timely implementation of policy and practice to prevent, rather than mitigate, further pollinator declines.

Methods.Using a horizon scanning approach we identified issues that are likely to impact pollinators, either positively or negatively, over the coming three decades.

Results.Our analysis highlights six high priority, and nine secondary issues. High priorities are: (1) corporate control of global agriculture, (2) novel systemic pesticides, (3) novel RNA viruses, (4) the development of new managed pollinators, (5) more frequent heatwaves and drought under climate change, and (6) the potential positive impact of reduced chemical use on pollinators in non-agricultural settings.

Discussion. While current pollinator management approaches are largely driven by mitigating past impacts, we present opportunities for pre-emptive practice, legislation, and policy to sustainably manage pollinators for future generations.

Introduction

Pollinators provide the key ecosystem service of pollination to agricultural crops and wild plants, with 35% of global crop production relying to some degree on pollination (Klein et al., 2007), along with more than 85% of wild flowering plants (Ollerton, Winfree & Tarrant, 2011). Consequently, declines in pollinators, which are occurring across the globe (Potts et al., 2010), may pose a significant threat to human and natural well-being. A suite of drivers, including habitat loss and homogenization (Kennedy et al., 2013), pesticides (Godfray et al., 2015), parasites and pathogens (e.g., Fürst et al., 2014; McMahon et al., 2015; Wilfert et al., 2016), invasive species (Stout & Morales, 2009), and climate change (e.g., Kerr et al., 2015) have been identified as past and current threats to pollinators (Vanbergen & The Insect Pollinator Initiative, 2013). Some actions to mitigate these threats, e.g., agri-environment schemes that provide forage and nesting resources (Batáry et al., 2015) and pesticide-use moratoriums to mitigate the potential impact of pesticides (Dicks, 2013), exist, but they have largely been applied post-hoc. While there is some evidence that such approaches might be mitigating pollinator losses (e.g., Carvalheiro et al., 2013), future sustainability of pollinators and the service they provide requires anticipation of potential threats and opportunities before they occur, enabling timely implementation of policy and practice to prevent, rather than mitigate, further pollinator declines.

One approach that can be used to anticipate future threats and opportunities for pollinators is the process of horizon scanning. Horizon scanning, a systematic technique to identify future threats or opportunities, is an important policy tool used in government and business to manage and proactively respond to upcoming threats and opportunities (Cook et al., 2014). In the last decade, horizon scanning has increasingly been applied to support environmental decision-making and inform policy and research on specific issues such as invasive species risk (Roy et al., 2014), management of particular geographic regions (Kennicutt et al., 2014) or threats to particular taxa (Fox et al., 2015). Proactive responses that pre-empt environmental risks are likely to be cheaper in the long term than reactive responses (e.g., Drechsler, Eppink & Wätzold, 2011) and potentially enable avoidance of substantial costs (Hulme et al., 2009).

Pollinator decline is one of the highest profile global environmental issues of the 21st century, as demonstrated through its selection by the International Platform on Biodiversity and Ecosystem Services (IPBES) as the subject of its first major assessment report (Gilbert, 2014). With governments around the world focused on this issue, and several producing national policies which largely focus around past and current threats, it is timely to identify forthcoming impacts on pollinators, both positive and negative, which may not yet be fully recognised by policy or research. Here we used a global horizon scanning team to identify potential future threats and opportunities for pollinators.

Methods

We followed a Horizon Scanning approach based on the Delphi method (Sutherland et al., 2016). The same approach has been used since 2010 to generate global horizon scans for conservation (Sutherland et al., 2016), and thus it provides a reliable and accepted methodology. The exercise was carried out by a core group of 17 pollinator experts (the authors), balanced across area of expertise and geographic knowledge. Experts were drawn from NGOs, research institutes, and universities. One member from the agrochemical industry accepted, but withdrew before the first stage of the process (see below) was completed. Table 1 shows how the group maps on to the two criteria of expertise and geography, and demonstrates strong coverage within the horizon-scanning group.

Table 1 The horizon-scanning group members were chosen to map across areas of research expertise and geographical knowledge.

Filled in cells in the table demonstrate this mapping.

	Africa	America	Asia	Australasia	Europe	
Agriculture						
Climate change						
Conservation						
Managed bees						
Other pollinators						
Pathogens						
Pollination						
Wild bees						

Selecting issues

Each person in the team consulted their networks and collected up to five potential horizon issues for consideration; 55 people (see ‘Acknowledgements’), in addition to the 17 experts, were consulted during this process. We searched for issues that were poorly known and considered likely to have a substantial impact on wild or managed pollinators (including insects, birds, mammals, and reptiles), either positive or negative, during the next one to 30 years. A ‘substantial’ impact could have a high magnitude, or take place over a large area, or both.

A long list of 60 issues, with associated references, was compiled (Table 2, Table S1) and sent to all core participants for a first round of anonymous scoring. Where the same issues had been identified by more than one member of the core group, these issues were grouped as one. Participants scored each issue from 1 (well known, unlikely to have substantial impact on pollinators) to 1,000 (poorly known, very likely to have substantial impact on pollinators). From these scores, we produced a ranked list of topics for each participant (the highest scored issue was given a rank of 1), and calculated the median rank for each topic (Table 2). Each person also stated whether they had previously heard of each issue or not.

Table 2 The results of the first round of voting on the horizon-scanning issues.

Each issue is listed with its median rank (low rank = most strongly voted for as a horizon issue) and its originality score (0 = not heard of, 1 = completely familiar)(see Methods for details). The number in the left column is simply the order in which issues were compiled.

#	Title	Median rank	Originality value	
1	Sulfoximine, a novel systemic class of insecticides	2	0.71	
2	The effect of chemical use on pollinators in non-agricultural settings	15	0.94	
3	Increasing use of fungicides	24	1.00	
4	Aluminium	44	0.29	
5	Potential non-target effects of nanoparticle pesticides on crop visiting insect pollinators	22	0.53	
6	Below-ground effects on plant–pollinator interactions	26	0.41	
7	Diffuse pollution: overlooked and underestimated?	27	0.47	
8	Policy and market factors exacerbate simplification of agricultural landscapes	15	0.94	
9	Soybean crop expansion worldwide	36	0.29	
10	Reduction or even removal of glyphosate	39	0.53	
11	Potential loss of floral resources for pollinators within and adjacent to agricultural lands through adoption of forthcoming ‘next generation’ genetically engineered crops and associated herbicide use	11	0.76	
12	Agricultural policy leading to intensification/abandonment/reforestation	35	1.00	
13	Land sparing (setting aside land for biodiversity conservation and intensifying production on remaining land)	27	0.88	
14	Lack of investment in research into sustainable farming methods	29	0.94	
15	Risks and opportunities of cutting pollinators out of food production	7	0.82	
16	Precision agriculture could improve pollination & reduce harm to pollinators	33	0.47	
17	Corporate farming could see effective alternative pollination systems adopted rapidly	33	0.53	
18	New positions open for alternative pollinators: must have good credentials	21	0.82	
19	Possible horticultural industry responses to pollinator limitation: bees in boxes	39	0.71	
20	GMO honey bees: a boon to pollination	33	0.35	
21	Natural selection and apiculture: breeding	42	0.82	
22	Entomovectoring	34	0.76	
23	Reduced budgets for public greenspace management	34	0.65	
24	Green roofs as potential pollinator habitat	40	0.82	
25	Climate change causing changes in crop distribution, leading to changes in managed pollinator distributions	31	0.59	
26	Socioeconomic drivers of change in flowering crops: unpredictable outcomes	24	0.76	
27	Benefits to pollinators from water quality protection	24	0.41	
28	Treatments for managed honeybee bacterial diseases using phage therapy	32	0.24	
29	Novel pathogens: a threat to many bee species and pollination	19	0.82	
30	Pollinators as pathways for pathogens	21	0.88	
31	Reductions in pollinator species richness may drive epidemics	15	0.29	
32	Honeybee viruses	36	1.00	
33	Bacterial diseases: American foulbrood & European foulbrood	53	0.94	
34	New emerging diseases: small hive beetle Aethina tumida	39	0.88	
35	New emerging diseases: Tropilaelaps spp.	29	0.53	
36	Varroa 2.0	28	0.41	
37	Infection with Nosema spp.	41	0.71	
38	Co-exposure between pesticides and pathogens	22	1.00	
39	Sanitary and genetic issues raised by international trade and globalization	21	1.00	
40	Climate change: altering pathogen epidemiology to the detriment of pollinators	15	0.59	
41	Changes in nutritional value of plants as a consequence of elevated atmospheric CO2 and pollution associated with human activities	19	0.41	
42	Increasing frequency of heatwaves and droughts may drive pollinator declines	15	0.88	
43	Impact of climate change on plant–pollinator interactions	24	0.88	
44	Impact of climate change on pollinator–pollinator interactions	30	0.47	
45	Decline and eventual disappearance of bumblebees due to climate change	38	0.94	
46	The impact of invasive alien commercial honeybees on native bees in Asia	17	0.76	
47	The spread of Apis cerana	33	0.53	
48	Use of managed bees to reduce human-wildlife conflict	42	0.59	
49	Substances that affect pollinator memory	36	0.82	
50	National and global monitoring: limited progress without them	24	0.88	
51	Altered evolutionary trajectories in plants and pollinators	22	0.47	
52	Environmental and ecological effect of Dams	51	0.50	
53	The bee band-wagon	24	0.65	
54	The media	43	0.82	
55	Focus on technology and commercialisation in science funding	24	0.82	
56	Destruction of roosting sites for pollinating bats worldwide	18	0.41	
57	Reproductive division of labor and susceptibility to stressors	45	0.59	
58	Gene drive technology to eradicate invasive pollinators	21	0.18	
59	Impacts of IPBES pollinators assessment	24	0.71	
60	Extinctions of flower-visiting birds	27	0.82	

Refining to a shortlist of priorities

The 28 issues with the lowest median ranks were retained, and participants had a chance to retain others they felt strongly should not be dismissed at this stage (no issues were brought back). Two participants were assigned to each of the 28 retained issues to research its technical details, likelihood, and potential impacts. These were not the same people who had suggested the issue.

Ten of the participants convened in Paola, Malta, in November 2015. We discussed each of the 28 issues in turn, with the constraint that the individual who suggested an issue was not the first to contribute to its discussion. All participants could see the median ranks and the percentage of the group who had heard of each issue (given as ‘originality value’ in Table 2), from round 1. Some issues were modified during this discussion. After each issue was discussed, participants independently and privately scored between 1 and 1,000 as previously described. The ‘originality value’ was used as a guide for scoring, although we were aware that, as the participants were all pollinator experts, it was unlikely to represent familiarity with these issues in the wider policy and research communities.

The remaining seven participants unable to attend the meeting took part in the process remotely, by submitting their research notes for issues they had been assigned (these were provided to each participant in printed form), and re-scoring independently after reading a detailed written account of the issues discussed.

The list of 15 issues presented here comprises those with the highest median ranks from the second round of scoring (Table 3). They are divided into High Priority and Secondary Priority issues (HPI, and SPI, respectively) because there was a clear break in the rankings among the top 15 issues, between the top six and the following nine. One issue (“Sanitary and genetic issues raised by international trade and globalization”) was removed from the final priority list despite having been ranked joint 13th by its median rank. While clearly important, the group agreed in the final stage that this was a current, well-known issue, and not an emerging issue on the horizon.

Table 3 The final results of the second round of voting on the reduced list of horizon-scanning issues.

Each issue is shown with its median rank. Note that the title of some issues were changed based on discussion prior to the second round of voting.

#	Title	Median rank	
1	Sulfoximine, a novel systemic class of insecticides	5	
2	Positive effects of reduced chemical use on pollinators in non-agricultural settings [new title]	7	
3	Increasing use of fungicides	12	
5	Potential non-target effects of nanoparticle pesticides on crop visiting insect pollinators	11	
6	Below-ground effects on plant–pollinator interactions	16	
8	Corporate control of agriculture at the global scale [new title]	4	
11	Potential loss of floral resources for pollinators within and adjacent to agricultural lands through adoption of forthcoming ‘next generation’ genetically engineered crops and associated herbicide use	16	
15	Risks and opportunities of cutting pollinators out of food production	12	
18	Increased diversity of managed pollinator species [new title]	6	
26	Socioeconomic drivers of change in flowering crops: unpredictable outcomes	20	
27	Benefits to pollinators from water quality protection	18	
29	Novel emerging RNA viruses [new title]	5	
30	Pollinators as pathways for pathogens	13	
31	Reductions in pollinator species richness may drive epidemics	13	
38	Co-exposure between pesticides and pathogens	22	
39	Sanitary and genetic issues raised by international trade and globalization	13	
40	Climate change: altering pathogen epidemiology to the detriment of pollinators	14	
41	Changes in nutritional value of plants as a consequence of elevated atmospheric CO2 and pollution associated with human activities	21	
42	Effects of extreme weather events under climate change [new title]	6	
43	Impact of climate change on plant–pollinator interactions	20	
46	The impact of non-native managed pollinators on native bee communities in Asia	13	
50	National and global monitoring: limited progress without them	19	
51	Altered evolutionary trajectories in plants and pollinators	25	
53	The bee band-wagon	26	
55	Focus on technology and commercialisation in science funding	23	
56	Destruction of bat roosts worldwide [new title]	15	
58	Gene drive technology to eradicate invasive pollinators	25	
59	Impacts of IPBES pollinators assessment	12	

Results

Using a modified Delphi process, we identified 60 initial issues of interest (Table 2, Table S1), which reduced to six high priority issues and nine secondary priority issues (Table 3). These issues can be partially mapped onto areas previously identified as being important causes of pollinator decline, e.g., agricultural practices (Fig. 1, Table 4). However, the issues we identified are largely distinct from past and current drivers of pollinator abundance, and require distinct policy and practices to minimize the threat and maximise the opportunities they present (Table 4). As is standard for a horizon scanning process, the identified issues are presented in rank order below, with the highest ranked issue first.

Figure 1 A schematic showing how the horizon scanning issues for pollinators map onto existing known drivers of pollinator decline, following Vanbergen & The Insect Pollinator Initiative (2013), and novel drivers with positive or negative opportunities.

Table 4 The relationship between horizon scanning issues, past problems and actions, and future responses.

The relationship between responses to current or past issues (column 1), identified horizon issues grouped by overarching driver (column 2), and potential pro-active responses to these issues (column 3).

Current responses, suggested or enacted, to related non-horizon issues	Horizon issues	Potential responses to horizon issues	
Habitat loss & homogenisation	HPI-1, SPI-9		
Agri-environmental schemes; paying farmers to cover the costs of pollinator conservation measures so as to connect habitat patches to allow pollinator movement	Corporate control of agriculture at global scale	Consumer-led certification schemes focused on pollinators	
	Corporate Social Responsibility commitments to pollinators (or wider biodiversity)	
Habitat protection	Destruction of bat roosts	Legal protection of bat roosts as sanctuaries, especially in the tropics	
		Education of land owners about bat conservation	
		Research to assess the impact of bat declines on pollination services	
Pesticides	HPI-2, HPI-6, SPI-1, SPI-2		
Pesticide risk assessment and regulation	Sulfoximine pesticides	Pesticide risk assessment and regulation urgently needs to incorporate chronic, sub-lethal, indirect, and interactive impacts and in-field realistic trials using a range of pollinator species	
Reduce pesticide use (for example, through Integrated Pest Management)		
Reduced exposure through technological inovation (e.g., minimise spray dust and drift)	Reduced impacts in non-agricultural settings	Monitor impacts of pesticide use in non-agricultural setting	
	Nanoparticle pesticides	Research into impacts of nanoparticles on pollinators	
	Increasing fungicide use	Global and national campaigns to reduce and replace chemical usage in urban and suburban areas	
Parasites & Pathogens	HPI-3, SPI-5, SPI-6		
The World Organization for Animal health (OIE http://www.oie.int) regulations for transport and screening of bees	New RNA viruses	A coordinated international network for detecting the emergence of viral diseases of managed pollinators	
Reduced pollinator richness drives epidemics	
Pollinators as disease vectors	Consider methods of pollinator management in plant disease control	
Climate change	HPI-5, SPI-8		
Connect habitat patches to allow pollinator movement	Effects of extreme weather events	Targeted measures to reduce impacts of extreme temperatures, rainfall or drought (e.g., planting flower strips with drought resistant flower species)	
Diversify farming practices, such as through crop rotation, to reduce risk		
		Develop and use alternative climate resilient managed pollinator species	
	Altered pathogen epidemiology	Predict changes in distribution of pathogens under climate change	
Invasive Species	SPI-7		
Listing potentially invasive species	Invasive bees in Asia	Prevent or regulate use of non-native managed bee species, especially Bombus terrestris, which is known to be invasive	
Biosecurity measures		
Regulations on international trade and movements		Surveillance in at risk areas	
Novel Areas:			
	Increased diversity of managed pollinators (HPI-4)	Identify candidate wild pollinators for management	
		Risk assessment and regulation of movement around deployment of new managed pollinator species	
	Cutting pollinators out of food production (SPI-3)	Re-calibrate conservation to recognise the inherent value of pollinators, outside food production	
		Quantify range of risks and benefits to sustainable food production	
	Impacts of IPBES pollinators assessment (SPI-4)	Incorporate outputs into national and international policies relevant to pollinators including agriculture, pesticide, conservation and planning sectors	

HPI-1: corporate control of agriculture at the global scale

Consolidation in agri-food industries has led to unprecedented control over land access, land use and agricultural practices by a small number of companies (Worldwatch Institute, 2013). A newer trend is transnational land deals for crop production, which now occupy over 40 million hectares (http://www.landmatrix.org/en/), including areas of Brazil for soybean export to China, and West Africa for rubber and palm oil. Agri-food industries operating at scale tend to promote homogeneous production systems, which is rapidly changing landscapes, especially in the southern hemisphere (Laurance et al., 2014) in a way that could substantially reduce the diversity and abundance of native pollinators. From an opportunity perspective, large-scale control over agricultural practices could, under appropriate management practices, enable sustainable pollinator management to optimize pollination with respect to consumer demands.

HPI-2: sulfoximine, a novel systemic class of insecticides

Sulfoximines are a new class of insecticide that resemble neonicotinoids in mode of action, yet differ sufficiently to prevent cross-resistance (Sparks et al., 2013). The first sulfoximine to be marketed is Sulfoxaflor. In spray formulation, it is rapidly being registered for widespread crop use in countries across the globe, to combat rising resistance to neonicotinoids (Bass et al., 2015). If, as is likely, sulfoximines are next registered as seed treatments, they may soon replace neonicotinoids over vast geographic areas (Simon-Delso et al., 2015). Neonicotinoids have sub-lethal effects on wild pollinators (e.g., Rundlöff et al., 2015), which may be generated through impacts on neural processes and immunity (e.g., Di Prisco et al., 2013), but those of sulfoximines have not been studied. Seed treatments are particularly likely to generate sub-lethal effects broadly, since they are applied prophylactically, rather than sprayed at specific times (where usage may be modified to reduce or avoid impacts on pollinators). Thus, the rapid proliferation of a new systemic, neuroactive insecticide without sufficient testing for sub-lethal effects is a grave concern, particularly if new formulations such as seed treatments arise.

HPI-3: new emerging RNA viruses

Emerging infectious diseases–some transmitted by exotic ectoparasitic Varroa destructor mites–are considered major causes of colony decline for the most abundant commercial pollinator, the Western honey bee (Apis mellifera). Such diseases are shared with, and likely spill over into, wild pollinators (Fürst et al., 2014). Chief among them are RNA viruses, whose high mutation and recombination rates make them particularly likely to cross host backgrounds (Manley, Boots & Wilfert, 2015). There is substantial risk of novel viral diseases emerging with elevated virulence, more efficient transmission and broad host range. The threat to both wild and managed pollinators is exacerbated by transport of managed pollinators to new locations, which may bring RNA viruses into contact with novel vectors (Roberts, Anderson & Tay, 2015).

HPI-4: increased diversity of managed pollinator species

Managed pollinators can replace or augment wild pollinators, but currently very few species are employed—most commonly Apis mellifera and, to a lesser extent, some bumblebees, stingless bees, and solitary bees (Free, 1993; Delaplane & Mayer, 2000). Diversifying the species managed for pollination could enhance pollination in crops that either require specialist pollinators or do not receive optimal service from existing managed species; provide insurance against perturbations in the supply of existing species; and enable use of native species in regions where existing managed species are not native. It also represents a business opportunity. Developing alternative managed pollinators requires biological and technical knowledge about the focal species, to ensure reliable supplies for growers. Risks associated with deploying new species, including parasite transmission, competition with local pollinators, introgression with the local gene pool, and ecosystem level impacts (Stout & Morales, 2009), require proactive risk assessment and regulation.

HPI-5: effects of extreme weather events under climate change

Effects of gradually changing climate on pollinators are increasingly well characterised, while the impacts of extreme events are poorly understood. Projected increases in frequency, magnitude, or intensity of, e.g., heatwaves and droughts are very likely across substantial parts of the globe (IPCC Summary for Policymakers, 2013). Heatwaves and droughts can affect pollinators directly, or indirectly by generating resource bottlenecks (Takkis et al., 2015). There is evidence that such weather patterns can lead to local extinction of pollinators (Rasmont & Iserbyt, 2012; Oliver et al., 2015) potentially leading to the breakdown of plant–pollinator relationships (Harrison, 2000). Greater knowledge of the relative importance of different extreme events is urgently needed to future-proof pollinator-friendly habitat management.

HPI-6: positive effects of reduced chemical use on pollinators in non-agricultural settings

Chemicals that have negative impacts on pollinators are widely used in urban and suburban areas, and in the wider landscape (e.g., golf courses). Recent recognition of the value of such areas for pollinators (Baldock et al., 2015) provides an opportunity to increase awareness of chemical use, and drive successful ‘reduce and replace’ campaigns. The potential for large-scale reduction in chemical use across ever-growing urban and suburban areas could have significant positive impacts on insect pollinators (Muratet & Fontaine, 2015).

SPI-1: potential non-target effects of nanoparticle pesticides on crop visiting insect pollinators

Nanoparticle pesticide use is rapidly expanding (Sekhon, 2014), yet non-target effects have not been evaluated, and this technology may evade existing pesticide regulatory processes. Though major knowledge gaps exist, nanoparticle pesticides may adversely affect crop-visiting pollinators.

SPI-2: increasing use of fungicides

Fungicide use is expected to increase with higher summer rainfall, which has been predicted for many regions under climate change scenarios (IPCC Summary for Policymakers, 2013). Current risk assessments for fungicides fail to capture sub-lethal and indirect impacts (e.g., on bee gut flora and fungi in pollen stores, synergies between fungicides and insecticides, and elevated susceptibility to disease (Pettis et al., 2013)).

SPI-3: risks and opportunities of cutting pollinators out of food production

Plant breeding technology can produce crop varieties that do not require biotic pollination (Mazzucato et al., 2015). Wide uptake of this technology could stabilize yields and reduce costs, but could further entrench the pollinator crisis by removing the imperative for pollinator protection and threatening the viability of remaining pollinator-dependent crops.

SPI-4: impacts of IPBES pollinators assessment

The Intergovernmental Platform on Biodiversity and Ecosystem Services’ 2016 global assessment “Pollinators, Pollination and Food Production” (IPBES, 2016) is a critical evaluation of evidence on the status, value and threats to pollinators and pollination worldwide. It could galvanise or inform substantial new actions by governments, practitioners and researchers.

SPI-5: pollinators as pathways for pathogens

While visiting flowers, pollinators can also transmit plant and pollinator diseases (McArt et al., 2014). Crop industries concerned about pollinator-mediated disease spread could enact restrictions on movements of managed pollinators, providing economic incentive to prioritise the use of local wild pollinators.

SPI-6: reductions in pollinator species richness may drive epidemics

Infectious disease transmission involves interactions among networks of species. The inverse relationship between host species diversity and disease transmission (Civitello et al., 2015) could drive disease epidemics as pollinator diversity declines.

SPI-7: the impact of non-native managed pollinators on native bee communities in Asia

The commercial importation of European Bombus terrestris (He et al., 2013) is very likely to negatively impact bumblebee communities in China, the global centre of bumblebee species diversity, as it has in other areas (e.g., Morales et al., 2013). The eight native honey bee species are increasingly likely to be negatively impacted by commercial import of A. mellifera and other managed bees.

SPI-8: climate change: altering pathogen epidemiology to the detriment of pollinators

In addition to direct and indirect impacts on pollinators, climate change may alter pollinator susceptibility to disease or enhance environmental transmission of pathogens (Natsopoulou et al., 2015). This may change pathogen range, prevalence, epidemiology, and the impact of emerging infectious disease agents on pollinators and pollination.

SPI-9: destruction of bat roosts worldwide

Globally, bats face increasing threats (Regan et al., 2015) due to habitat loss, roost destruction, hunting and persecution. As human activities expand into tropical forest areas, destruction of roost sites will increase, while culling is an increasing threat. Bats are important pollinators in tropical forests, savannas, deserts, and for cultivated plants (e.g., agave). The consequences of precipitous declines in bat pollination have not been assessed.

Discussion

Here we have identified a series of horizon issues, both positive and negative, for pollinators. Interestingly, while some of these have connections to previous causes of pollinator declines, and can be linked to over-arching drivers, such as agriculture and climate change, the policy and practice needed to minimize future threats and maximise future opportunities are largely distinct from current best practice in pollinator conservation.

In addition to their direct effects, the horizon issues identified in this study may also interact to positively and negatively impact pollinators. For example, extreme weather events driven by climate change are likely to influence corporate agriculture, its location, and its spread across the globe, whilst at the same time calling for agricultural practices that develop or support locally specialized pollinators. Such interactions deserve further investigation.

Horizon-scanning projects are, of necessity, limited by the panel make-up and the range of sources they can draw on. We specifically invited panel members from all major geographical regions, and across government research institutes, industry, NGOs, and universities, in order to maximise the breadth of knowledge and experience in our panel. To increase this breadth even further, panel members consulted a wide range of experts. Nevertheless, we acknowledge that an alternative panel make-up could have arrived at a different ordering, or selection of issues. In addition, our selection of issues should not be taken as static. Horizon scanning detects possible future changes about which there is little current evidence (sometimes known as ‘weak signals’; Cook et al., 2014). As the future unfolds, new technologies and global change phenomena arise, and so the process should be repeated as an ongoing part of policy and research planning.

Future-proofing pollinators is urgently required, in a world where demand for pollination services is rising at the same time as threats are increasing (Lautenbach et al., 2012; Potts et al., 2010; Vanbergen & The Insect Pollinator Initiative, 2013). Many of the issues we identified are new developments relating to current problems for pollinators, but some are potential opportunities, or entirely new potential threats (Fig. 1). As indicated in Table 3, for some issues the appropriate policy responses or actions to mitigate negative impacts might be different from those currently discussed or enacted. For example, methods of pollinator management may be needed to control the spread of both plant and insect diseases in future, especially if the number of managed pollinator species, and the distances they are moved, increases. Legislation for pesticide development urgently needs to incorporate chronic and interactive impacts and proper field trials for future pesticides. Early identification of such issues provides the opportunity to develop policies and practices to limit negative impacts, or to take advantage of potential positive impacts (Table 3).

While all horizon-scanning exercises are limited in their outputs, we believe we have identified current key issues that should be the focus of conservation practitioners, industry, and policy-makers if we are to maintain and benefit from a functional pollinator assemblage at the global scale in the ensuing decades.

Supplemental Information

Table S1 The initial list of potential issues identified by the horizon-scanning process. Issues with the same number were grouped together, based on similarity, for voting

Click here for additional data file.

We would like to thank the following individuals, who were consulted during the first stage of the Horizon Scanning process: C Poole (SANBI, South Africa); P Towers, M Ishii-Eitemann, and E Marquez (Pesticide Action Network, USA); E Mader, A Code, M Vaughan, and S Black (Xerces Society, USA); M Goodwin, P Schaare, L Evans, B Howlett, and C Hall (Plant & Food Research, New Zealand); T Breeze, R Girling, J Wickens, V Wickens, and J Bishop (University of Reading, UK); V Doublet (University of Exeter, UK); M Husemann, P Theodorou, R Moritz, M Natsopoulou, and A Soro (Halle University, Germany); S Jha (UT Austin, USA); M Lopez-Uribe (North Carolina State University, USA); J Lozier (University of Alabama, USA); G Suwannapong (Burapha University, Thailand); A Zayed (York University, Canada); A Bezerra (Universidade Federal do Ceará, Brazil); V Imperatriz-Fonseca (University of Sao Paolo, Brazil); B Blochtein (Pontifical Catholic University of Rio Grande do Sul, Brazil); S Wiantoro (Museum Zoologicum Bogoriense, Indonesia); J Ollerton (University of Northampton); M Goddard (University of Newcastle); T Ahjohkoski (Bristol City Council); J Zimmermann, E Power, F Hecq, A Delaney, C Owens, and S Kavanagh (Trinity College Dublin, Ireland); M Coulon, E Dubois, V Duquesne, S Franco, M-P Rivière (ANSES, France); F Mutinelli (IZSV, Italy); Brandom Keim (USA); Becky LeAnstey (Environment Agency, UK); A Harmon-Threatt (University of Illinois, USA); Q McFrederick (UC Riverside, USA); M Spivak (University of Minnesota, USA); A-M Klein (University of Freiburg, Germany); M Fox (Environment Agency, UK).

Additional Information and Declarations

Competing Interests

Author Contributions

Data Availability

The authors declare there are no competing interests.

Mark J.F. Brown, Lynn V. Dicks and Robert J. Paxton conceived and designed the experiments, performed the experiments, analyzed the data, wrote the paper, prepared figures and/or tables, reviewed drafts of the paper.

Katherine C.R. Baldock, Andrew B. Barron, Marie-Pierre Chauzat, Breno M. Freitas, Dave Goulson, Sarina Jepsen, Claire Kremen, Jilian Li, Peter Neumann, Simon G. Potts, Colleen L. Seymour and Jane C. Stout performed the experiments, analyzed the data, wrote the paper, reviewed drafts of the paper.

David E. Pattemore and Oliver Schweiger performed the experiments, analyzed the data, wrote the paper, prepared figures and/or tables, reviewed drafts of the paper.

The following information was supplied regarding data availability:

The raw data has been supplied as Table S1.

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
