# Peer review of "A horizon scan of future threats and opportunities for pollinators and pollination"

_PeerJ, doi:10.7717/peerj.2249_

## Round 0.1 · original submission · Major Revisions

Three reviewers have commented on your manuscript. As you will see all of them are positive and their comments will improve the manuscript. Please revise and resubmit in due course.

·

Basic reporting

The article addresses an interesting subject. The problem is well presented and the results are clearly discussed.
I would suggest a better discussion of two points (see general comments for authors).

Experimental design

The methods used by the authors are standard methods and have been applied properly.
In the general comments for authors I suggest to expand the description of the very first phase of the study.

Validity of the findings

See Comments for the Author

Additional comments

The article reports the results of an interesting activity carried out to identify issues that will likely impact pollinators over the coming three decades. The work is justified by the need of anticipating potential threats and opportunities before they occur and was carried out using a standard method.

I confess that I’ve got no experience on procedures such as the horizon scan exploited for preparing the submitted manuscript. However, it seems to me that, regardless of the method used for extracting the best information from the panel of experts, that is certainly appropriate and I’m not questioning, the possible outcome clearly depends on the group of people taking part in the activity (lines 296-305 show that authors are well aware of the problem).
For this reason I believe that some more words should be dedicated to this aspect (at present there are just four rows), showing how the group of people involved in the project was actually suitable for the task.
I also think that a figure or a table should be added to the paper synthesizing in an intelligible way this important aspect.
Following is a suggestion.
It seems from the words of the authors that the “area of expertise” (e.g. domestic bees, wild bees, bee pathology, apiculture, pollination, biodiversity, etc.) and “geographical region” (e.g. North and South America, Europe, Africa, etc.) are relevant factors for the selection of the experts and must be taken into account and well balanced.
Basically, in order to get the most comprehensive representation of the subject, including all relevant aspects and “global” in its nature, the experts should cover at best the virtual space of the possible intersections between area of expertise and geographic region.
I suspect that some method already exists for illustrating this but, if I had to imagine one, I’d think about a rectangular matrix with as many columns (or rows) as the geographical regions and a number rows (or columns) corresponding to the areas of expertise. Squares in the grid should be filled with the initials of the experts.
In such a graphical representation, a well balanced group of experts would be the one covering all the space, with a rather homogeneous distribution. If a touch of statistics was needed, the uniformity of the distribution could be tested by looking at the variance/mean ratio of the number of experts per square, that should be lower or similar to one. A ratio higher than one would mean a clumped distribution with overrepresented areas of origin or expertise as well as some unrepresented areas.

Following are some more observations related to single points. Of course, the list of priority issues is the result of the work and can not be questioned but there are a few clarifications that could added to the manuscript to highlight the importance of some of the issues.

1. Introduction (lines 60-64): in the list of the drivers of decline, many of the cited papers do not describe general effects but pertain to the effects of a subsample of each driver on a subsample of pollinators, for example: wild pollinators rather than bees, neonicotinoids rather than pesticides, Varroa and DWV rather than pathogens; I’d suggest to add an “e.g.” between brackets, before the reference.

2. McMahon et al., 2015 is not in the reference list

3. Line 141: add HP and SP in brackets after mentioning high priority and secondary priority issues.

4. HPI-2: two interesting points are discussed by the authors in this respect.
a. Authors say that sulfoximine insecticides are likely to generate sublethal effects because they resemble neonicotinoids, that are neuroactive, and actually cause sublethal effects.
This is well in line with a first line of evidence connecting neonicotinoids to colony losses in relation to their effect on the homing ability of foragers, however, there is another line of evidence pointing to another effect of such chemicals, that is related to the interplay between nervous system and immunity and the possible consequences for the balance between microbes and bees (see Di Prisco et al., 2013). Nobody knows yet if sulfoximine can do the same but the fact that they are nicotinic acetylcholine receptor (nAChR) competitive modulators as well as neonicotinoids is a matter of concern that should be considered.
b. I wonder if the fact that these chemicals are systemic and are thus applied prophylactically should be stressed a little bit more in relation to the fact that at present, the coexistence between bees and insecticides in many countries is based on the ban of insecticide use during flowering, a possibility clearly excluded in case of systemic insecticides.

5. Line 173: sulfoxaflor

6. HPI-3: I think that authors should better explain the link between high mutation and recombination rate and the emergence of novel viral diseases adding a couple of references (I think: Ryabov et al., on PLoS Path for the recombination issue and Wilfert et al., on Science for the mutation issue may be appropriate).

7. Line 280: there is a dot after e.g.

8. Table 1 is redundant in my opinion.

9. Line 320: “limited in scope”: I may not understand the exact meaning of this sentence but I would think that the possible results of the procedure rather than the scope are limited; to me the scope (identifying issues likely impacting pollinators over the coming three decades) is quite ambitious.

Reviewer 2 ·

Basic reporting

The English is fine but the text is highly ambiguous and methods are very unclear.
Relevant literature is cited. This study is a result of survey work through a process of Horizontal Scanning, so does not follow typical format for a research article based on experimental findings. The horizontal scanning method is interesting and seems potentially appropriate way to identify future positive and negative impacts on pollinators, to raise awareness to policy makers, researchers, action groups and the public.

Experimental design

I am completely confused about the methods:
First, the authors “searched for issues that were poorly known and likely to have an impact” on pollinators. (line 105). It sounds like a conundrum: If issues are poorly known, how would anyone know if they are likely to have an impact?

Line 113: issues were scored from 1 = well known OR? unlikely to have impact. The word “or” is ambiguous (either of those, or that is, unlikely…)

Then top score was 1000. But do they mean 100? The top score would be poorly known, very likely to have impact

Then they calculated median rank for first round of survey results in Table 1 and retained the 28 “highest” median ranks, but the legend for Table 1 says lowest median is most likely to have an impact. I got lost at this point. In Table 1, the numbers in left hand column presumably are the issues? Should be labeled in legend.

On Line 120, they say thy picked out 28 with "highest" median ranks (I think they really picked the lowest median ranks), but what happened to originality scores? I’m unclear what the originality score is and how it was used.

For the second round of scoring, line 141 - 150, they say there was a clear break between HP1 and SP1 and refer to Table 2, but I don’t see the clear break.

Table 3: has both HP1 and SP1 numbers in Horizon issues column. Why?

I think the findings are interesting but I got lost in methods and stopped reading because I don’t know how to interpret them.

Validity of the findings

Completely, and unfortuntely, unclear

Additional comments

Please see comments above in Experimental Design. More explanation and clarity is needed to understand how the Horizontal Scanning was done. Table 3 is very unclear.

Reviewer 3 ·

Basic reporting

no comments

Experimental design

no comments

Validity of the findings

no comments

Additional comments

In the present manuscript, the Authors aim to prevent of potential threats before they occurs, in order to apply a correct decision making in the policy pratics. This is the core of "Horizon Scan Approach" (HSA), a promising methods for determining problems and opportunities through a systematic examination of potential threats, in this case for pollinators and pollination management.


The topic is interesting and the manuscript well conceived and writed, however HSA is not lacking of limiting/misleading factors. The main one, as the Authors suggested in the discussion dection (296-305) is the limited number of panel make-up members; this should tend (theoretically) to infinity, with a early correction mechanism to prevent the eventual wrong course of the HSA process in order to test and increase the results robustness.

In my opinion, HSA method should be open to a number as large as possible of qualified people with an internal correction mechanism to be implemented with other methods of future scenarios prediction, continuing (in parallel) the mitigation of past impacts.

I think that by presenting the HSA metod not as the only and decisive right approach there will be a good chance to be published soon.

---

## Round 0.2 · Minor Revisions

Some final minor revisions are required before acceptance.

·

Basic reporting

It seems to me that the authors have seriously considered the points raised by all reviewers and revised their manuscript accordingly.

Experimental design

no further comments

Validity of the findings

no further comments

Reviewer 3 ·

Basic reporting

no comment

Experimental design

no comment

Validity of the findings

no comment

Additional comments

no comment

Reviewer 4 ·

Basic reporting

No comments

Experimental design

No comments

Validity of the findings

No comments

Additional comments

This well-written manuscript addresses an important and forward-thinking aspect of pollinator decline: to anticipate threats to pollinators in the near future instead of passively waiting for their declines to try to mitigate existing problems.

I really enjoyed reading this manuscript and I think it provides valuable information that will focus future efforts to protect pollinators from population crashes. I only have a couple of very minor comments.

* This comment may be the result of my own biases while reading something regarding “pollinators”. My impression throughout this manuscript was that the authors were mainly referring to bees when talking about pollinators. However, in line 245 they mention “destruction of bat roots worldwide” as a secondary priority. This made me wonder if all pollinators were included in the horizon scan approach. If they were, it would be good to remind the readers who were the pollinators included in the analysis. It surprised me that there was no mention of specific needs for butterfly pollinators, a group of pollinators that is also currently threatened.

* L60 - I think “flowering Angiosperms” is redundant. Either “flowering plants” or “Angiosperms” would be ok.

---

## Round 0.3 · accepted · Accept

The manuscript has been improved according to the suggested minor revisions and can be accepted now.